# ON THE ADVERSARIAL ROBUSTNESS OF VISION TRANSFORMERS

## ABSTRACT

Following the success in advancing natural language processing and understanding, transformers are expected to bring revolutionary changes to computer vision. This work provides a comprehensive study on the robustness of vision transformers (ViTs) against adversarial perturbations. Tested on various white-box and transfer attack settings, we find that ViTs possess better adversarial robustness when compared with convolutional neural networks (CNNs). This observation also holds for certified robustness. We summarize the following main observations contributing to the improved robustness of ViTs: 1) Features learned by ViTs contain less low-level information and are more generalizable, which contributes to superior robustness against adversarial perturbations. 2) Introducing convolutional or tokens-to-token blocks for learning low-level features in ViTs can improve classification accuracy but at the cost of adversarial robustness. 3) Increasing the proportion of transformers in the model structure (when the model consists of both transformer and CNN blocks) leads to better robustness. But for a pure transformer model, simply increasing the size or adding layers cannot guarantee a similar effect. 4) Pre-training without adversarial training on larger datasets does not significantly improve adversarial robustness though it is critical for training ViTs. 5) Adversarial training is also applicable to ViT for training robust models. Furthermore, feature visualization and frequency analysis are conducted for explanation. The results show that ViTs are less sensitive to high-frequency perturbations than CNNs and there is a high correlation between how well the model learns low-level features and its robustness against different frequency-based perturbations.

## 1 INTRODUCTION

Transformers are originally applied in natural language processing (NLP) tasks as a type of deep neural network (DNN) mainly based on the self-attention mechanism (Vaswani et al. (2017); Devlin et al. (2018); Brown et al. (2020)), and transformers with large-scale pre-training have achieved state-of-the-art results on many NLP tasks (Devlin et al. (2018); Liu et al. (2019); Yang et al. (2019); Sun et al. (2019)). Recently, Dosovitskiy et al. (2020) applied a pure transformer directly to sequences of image patches (i.e., a vision transformer, ViT) and showed that the Transformer itself can be competitive with convolutional neural networks (CNN) on image classification tasks. Since then transformers have been extended to various vision tasks and show competitive or even better performance compared to CNNs and recurrent neural networks (RNNs) (Carion et al. (2020); Chen et al. (2020); Zhu et al. (2020)). While ViT and its variants hold promise toward a unified machine learning paradigm and architecture applicable to different data modalities, it remains unclear on the robustness of ViT against adversarial perturbations, which is critical for safe and reliable deployment of many real-world applications.

In this work, we examine the adversarial robustness of ViTs on image classification tasks and make comparisons with CNN baselines. As highlighted in Figure 1(a), our experimental results illustrate the superior robustness of ViTs than CNNs in both white-box and black-box attack settings, based on which we make the following important findings:

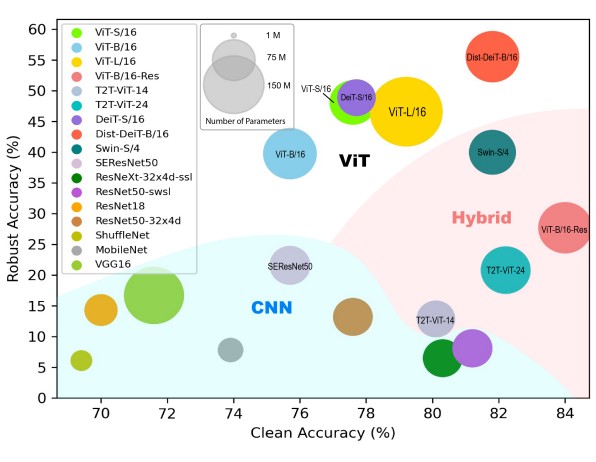
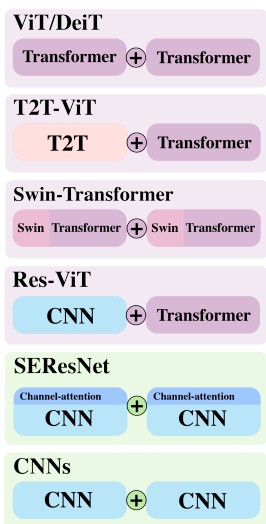

(a) Robust Accuracy v.s. Clean Accuracy

(b) Model Architecture

Figure 1: (a) Robust accuracy v.s. clean accuracy. The robust accuracy is evaluated by AutoAttack (Croce & Hein (2020)). The "Hybrid" class includes CNN-ViT, T2T-ViT and Swin-T as introduced in Section 3. Models with attention mechanisms have their names printed at the center of the circles. ViTs have the best robustness against adversarial perturbations. Introducing other modules to ViT can improve clean accuracy but hurt adversarial robustness. CNNs are more vulnerable to adversarial attacks. (b) Vision transformers and CNNs investigated in this paper.

- Features learned by ViTs contain less low-level information and benefit adversarial robustness. ViTs achieve a lower attack success rate (ASR) of $51.9\%$ compared with a minimum of $83.3\%$ by CNNs in Figure 1(a). They are also less sensitive to high-frequency adversarial perturbations.

- Using denoised randomized smoothing (Salman et al., 2020), ViTs attain significantly better certified robustness than CNNs.

- It takes the cost of adversarial robustness to improve the classification accuracy of ViTs by introducing blocks to help learn low-level features as shown in Figure 1(a).

- Increasing the proportion of transformer blocks in the model leads to better robustness when the model consists of both transformer and CNN blocks. For example, the attack success rate (ASR) decreases from $87.1\%$ to $79.2\%$ when $10$ additional transformer blocks are added to T2T-ViT-14. However, increasing the size of a pure transformer model cannot guarantee a similar effect, e.g., the robustness of ViT-S/16 is better than that of ViT-B/16 in Figure 1(a).

- Pre-training without adversarial training on larger datasets does not improve adversarial robustness though it is critical for training ViT.

- The principle of adversarial training through min-max optimization (Madry et al. (2017); Zhang et al. (2019)) can be applied to train robust ViTs.

## 2 RELATED WORK

Transformer (Vaswani et al. (2017)) has achieved remarkable performance on many NLP tasks, and its robustness has been studied on those NLP tasks. Hsieh et al. (2019); Jin et al. (2020); Shi & Huang (2020); Li et al. (2020); Garg & Ramakrishnan (2020); Yin et al. (2020) conducted adversarial attacks on transformers including pre-trained models, and in their experiments transformers usually show better robustness compared to other models based on Long short-term memory (LSTM) or CNN, with a theoretical explanation provided in Hsieh et al. (2019). However, due to the discrete nature of NLP models, these studies are focusing on discrete perturbations (e.g., word or character substitutions) which are very different from small and continuous perturbations in computer vision tasks. Besides, Wang et al. (2020a) improved the robustness of pre-trained transformers from an

information-theoretic perspective, and Shi et al. (2020); Ye et al. (2020); Xu et al. (2020) studied the robustness certification of transformer-based models. To the best of our knowledge, this work is one of the first studies that investigates the adversarial robustness (against small perturbations in the input pixel space) of transformers on computer vision tasks. There are some concurrent works studying the adversarial robustness of ViTs. We supplement their contribution in Appendix G.

## 3 MODEL ARCHITECTURES

We first review the architecture of models investigated in our experiments including several vision transformers (ViTs) and CNN models. A detailed table of comparison is given in Table 5.

### 3.1 VISION TRANSFORMERS

We consider the original ViT (Dosovitskiy et al. (2020)) and its four variants shown in Figure 1(b).

**Vision transformer (ViT) and data-efficient image transformer (DeiT):** ViT ( Dosovitskiy et al. (2020)) mostly follows the original design of Transformer (Vaswani et al. (2017); Devlin et al. (2018)) on language tasks. For a 2D image $x \in \mathbb{R}^{H \times W \times C}$ with resolution $H \times W$ and $C$ channels, it is divided into a sequence of $N = \frac{H \cdot W}{P^2}$ flattened 2D patches of size $P \times P$, $x_i \in \mathbb{R}^{N \times (P^2 \cdot C)}$ ($1 \leq i \leq N$). The patches are first encoded into patch embeddings with a simple convolutional layer, where the kernel size and stride of the convolution is exactly $P \times P$. In addition, there are also position embeddings to preserve positional information. Similar to BERT (Devlin et al. (2018)), a large-scale pre-trained model for NLP, a special `[CLS]` token is added to output features for classification. DeiT ( Touvron et al. (2021)) further improves the ViT's performance using data augmentation or distillation from CNN teachers with an additional distillation token. We investigate ViT-{S,B,L}/16, DeiT-S/16 and Dist-DeiT-B/16 as defined in the corresponding papers in the main text and discuss other structures in Appendix F.

**Hybrid of CNN and ViT (CNN-ViT):** Dosovitskiy et al. (2020) also proposed a hybrid architecture for ViTs by replacing raw image patches with patches extracted from a CNN feature map. This is equivalent to adding learned CNN blocks to the head of ViT as shown in Figure 1(b). Following Dosovitskiy et al. (2020), we investigate ViT-B/16-Res in our experiments, where the input sequence is obtained by flattening the spatial dimensions of the feature maps from ResNet50.

**Hybrid of T2T and ViT (T2T-ViT):** Yuan et al. (2021) proposed to overcome the limitations of the simple tokenization in ViTs, by progressively structurizing an image to tokens with a token-to-token (T2T) module, which recursively aggregates neighboring tokens into one token such that low-level structures can be better learned. T2T-ViT was shown to perform better than ViT when trained from scratch on a midsize dataset. We investigate T2T-ViT-14 and T2T-ViT-24 in our experiments.

**Hybrid of shifted windows and ViT (Swin-T):** Liu et al. (2021) computes the representations with shifted windows scheme, which brings greater efficiency by limiting self-attention computation to non-overlapping local windows while also allowing for cross-window connection. We investigate Swin-S/4 in the main text and discuss other structures in Appendix F.

### 3.2 CONVOLUTIONAL NEURAL NEWORKS

We study several CNN models for comparison, including ResNet18 (He et al. (2016)), ResNet50-32x4d (He et al. (2016)), ShuffleNet (Zhang et al. (2018)), MobileNet (Howard et al. (2017)) and VGG16 (Simonyan & Zisserman (2014)). We also consider the SEResNet50 model, which uses the Squeeze-and-Excitation (SE) block (Hu et al. (2018)) that applies attention to channel dimensions to fuse both spatial and channel-wise information within local receptive fields at each layer.

The aforementioned CNNs are all trained on ImageNet from scratch. For a better comparison with pre-trained transformers, we also consider two CNN models pre-trained on larger datasets: ResNeXt-32x4d-ssl (Yalniz et al. (2019)) pre-trained on YFCC100M (Thomee et al. (2015)), and ResNet50-swsl pre-trained on IG-1B-Targeted (Mahajan et al. (2018)) using semi-weakly supervised methods (Yalniz et al. (2019)). They are both fine-tuned on ImageNet.

# 4 ADVERSARIAL ROBUSTNESS EVALUATION METHODS

We consider the commonly used $\ell_\infty$-norm bounded adversarial attacks to evaluate the robustness of target models. An $\ell_\infty$ attack is usually formulated as solving a constrained optimization problem:

$$\max_{\mathbf{x}^{adv}} \mathcal{L}\left(\mathbf{x}^{adv}, y\right) \quad \text{s.t.} \quad \left\|\mathbf{x}^{adv} - \mathbf{x}_0\right\|_\infty \leq \epsilon, \tag{1}$$

where $\mathbf{x}_0$ is a clean example with label $y$, and we aim to find an adversarial example $\mathbf{x}^{adv}$ within an $\ell_\infty$ ball with radius $\epsilon$ centered at $\mathbf{x}_0$, such that the loss of the classifier $\mathcal{L}\left(\mathbf{x}^{adv}, y\right)$ is maximized. We consider untargeted attack in this paper, so an attack is successful if the perturbation successfully changes the model's prediction. The attacks as well as a randomized smoothing method used in this paper are listed below.

**White-box attack** Four white-box attacks are involved in our experiments. The Projected Gradient Decent (PGD) attack (Madry et al. (2017)) solves Eq. 1 by iteratively taking gradient ascent:

$$\mathbf{x}_{t+1}^{adv} = Clip_{\mathbf{x}_0, \epsilon}(\mathbf{x}_t^{adv} + \alpha \cdot \text{sgn}\left(\nabla_{\mathbf{x}} J\left(\mathbf{x}_t^{adv}, y\right)\right)), \tag{2}$$

where $\mathbf{x}_t^{adv}$ stands for the solution after $t$ iterations, and $Clip_{\mathbf{x}_0, \epsilon}(\cdot)$ denotes clipping the values to make each $\mathbf{x}_{t+1,i}^{adv}$ within $[\mathbf{x}_{0,i} - \epsilon, \mathbf{x}_{0,i} + \epsilon]$, according to the $\ell_\infty$ threat model. As a special case, Fast Gradient Sign Method (FGSM) (Goodfellow et al. (2014)) uses a single iteration with $t = 1$. AutoAttack (Croce & Hein (2020)) is currently the strongest white-box attack which evaluates adversarial robustness with a parameter-free ensemble of diverse attacks. We also design a frequency-based attack for analysis, which conducts attack under an additional frequency constraint:

$$\mathbf{x}_{freq}^{adv} = \text{IDCT}(\text{DCT}(\mathbf{x}_{pgd}^{adv} - \mathbf{x}_0) \odot \boldsymbol{M}_f) + \mathbf{x}_0, \tag{3}$$

where DCT and IDCT stand for discrete cosine transform and inverse discrete cosine transform respectively, $\mathbf{x}_{pgd}^{adv}$ stands for the adversarial example generated by PGD, and $\boldsymbol{M}_f$ stands for the mask metric defined the frequency filter which is illustrated in Appendix B. We found this design similar to Wang et al. (2020b).

**Black-box attack** We consider the transfer attack which studies whether an adversarial perturbation generated by attacking the *source* model can successfully fool the *target* model. This test not only evaluates the robustness of models under the black-box setting, but also becomes a sanity check for detecting the obfuscated gradient phenomenon (Athalye et al. (2018)). Previous works have demonstrated that single-step attacks like FGSM enjoys better transferability than multi-step attacks (Kurakin et al. (2017)). We thus use FGSM for transfer attack in our experiments.

**Denoised Randomized Smoothing** We also evaluate the certified robustness of the models using randomized smoothing, where the robustness is evaluated as the certified radius, and the model is certified to be robust with high probability for perturbations within the radius. We follow Salman et al. (2020) to train a DnCNN (Zhang et al. (2017)) denoiser $\mathcal{D}_\theta$ for each pre-trained model $f$ with the "stability" objective with $\mathcal{L}_{CE}$ denoting cross entropy and $\mathcal{N}$ denoting Gaussian distribution:

$$\mathcal{L}_{Stab} = \mathbb{E}_{(x_i, y_i) \in \mathcal{D}, \delta} \mathcal{L}_{CE}(f(\mathcal{D}_\theta(x_i + \delta)), f(x_i)) \text{ where } \delta \sim \mathcal{N}(0, \sigma^2 I). \tag{4}$$

Randomized smoothing is applied on the denoised classifier $f \circ \mathcal{D}_\theta$ for robustness certification:

$$g(x) = \arg\max_{c \in \mathcal{Y}} \mathbb{P}[f(\mathcal{D}_\theta(x + \delta)) = c] \text{ where } \delta \sim \mathcal{N}(0, \sigma^2 I). \tag{5}$$

The certified radius is then calculated for the smoothed classifier as (Cohen et al., 2019):

$$R = \frac{\sigma}{2}(\Phi^{-1}(p_A) - \Phi^{-1}(p_B)), \tag{6}$$

where $\Phi^{-1}$ is the inverse of the standard Gaussian CDF, $p_A = \mathbb{P}(f(x+\delta) = c_A)$ is the confidence of the top-1 predicted class $c_A$, and $p_B = \max_{c \neq c_A} \mathbb{P}(f(x + \delta) = c)$ is the confidence for the second top class. Accordingly, given a perturbation radius, the certified accuracy under this perturbation radius can be evaluated by comparing the given radius to the certified radius $R$.

## 5 EXPERIMENTAL RESULTS

In the experiments, we show the superior robustness of ViTs over CNNs against both white-box attacks and transfer attacks. We also study their certified robustness with the denoised randomized smoothing technique and conduct preliminary adversarial training experiments on ViT. Based on the experiments, we analyze what grants ViTs to be adversarially robust through different lens.

For all experiments, we load pre-trained ViT models and CNNs from the PyTorch image models library (timm, Wightman (2019)) and torchvision (Paszke et al. (2019)) respectively. We evaluate the clean accuracy of each model on the whole test set of ImageNet-1k (Deng et al. (2009)), and we sample 1,000 test examples to evaluate robust accuracy and attack success rate (ASR). Note that lower ASR means better robustness. Experimental results on CIFAR-10 are reported in Appendix E. For adversarial training we use CIFAR-10 (Krizhevsky et al. (2009)), as detailed in Section 5.4.

### 5.1 ROBUSTNESS UNDER WHITE-BOX ATTACKS

**Settings** We use PGD and AutoAttack to study the robustness under white-box attacks. We consider attack radius $\epsilon$ from $\{0.001, 0.003, 0.005, 0.01\}$. For PGD attack, we fix the attack steps to $n_{iter} = 40$ with other parameters following the default setting of the implementation in Foolbox (Rauber et al. (2020)). AutoAttack does not require any hyper-parameter tuning.

**Results** We present the results using PGD attack in Table 1 and the results using AutoAttack in Table 2. The ASR is approximately 100% on all the models when $\epsilon$ is large, e.g. when $\epsilon = 0.01$. But for smaller attack radii, ViT models have lower ASR than CNNs under both PGD attack and AutoAttack. For example, when $\epsilon = 0.001$, the ASR for ViT-S/16 is only 44.6% while the ASRs for CNNs are at least 70.0%. And under the same attack radius, the ASR of AutoAttack for ViT-S/16 is only 51.9% compared to 93.9% of ShuffleNet. Besides, the results show that AutoAttack is much stronger than PGD attack under the same $\epsilon$. These results demonstrate that ViT is more robust than CNNs under these white-box attacks. We also visualize the clean/robust accuracy tradeoff and model size of these models in Figure 1(a).

Table 1: Attack success rate (%) of target models against 40-step PGD attack with different radii and the clean accuracy ("clean acc").

Table 2: Attack success rate (%) of target models against AutoAttack with different attack radii, in a similar format as Table 1.

| Model | Clean Acc | PGD Attack radius | | | |
|---|---|---|---|---|---|
| | | 0.001 | 0.003 | 0.005 | 0.01 |
| ViT-S/16 | 77.6 | **44.6** | **75.4** | **89.8** | 99.0 |
| ViT-B/16 | 75.7 | 51.1 | 85.4 | 94.0 | 99.1 |
| ViT-L/16 | 79.2 | 44.9 | 76.6 | 90.1 | **98.2** |
| ViT-B/16-Res | 84.0 | 54.5 | 91.6 | 97.7 | 99.9 |
| T2T-ViT-14 | 80.1 | 62.9 | 93.0 | 98.2 | 100.0 |
| T2T-ViT-24 | 82.2 | 52.3 | 87.7 | 96.6 | 99.8 |
| Deit-S/16 | 77.7 | 51.1 | 82.4 | 92.9 | 98.9 |
| Dist-Deit-B/16 | 81.8 | 44.4 | 82.3 | 95.5 | 99.6 |
| Swin-S/4 | 81.8 | 60.0 | 87.6 | 96.8 | 99.8 |
| SEResNet50 | 75.7 | 64.6 | 95.1 | 99.2 | 99.9 |
| ResNeXt-32x4d-ssl | 80.3 | 77.0 | 97.1 | 98.8 | 99.5 |
| ResNet50-swsl | 81.2 | 75.3 | 97.1 | 98.6 | 99.6 |
| ResNet18 | 70.0 | 75.1 | 98.0 | 99.4 | 99.9 |
| ResNet50-32x4d | 77.6 | 71.8 | 96.8 | 98.8 | 99.6 |
| ShuffleNet | 69.4 | 85.0 | 99.4 | 99.8 | 100.0 |
| MobileNet | 71.9 | 83.3 | 99.6 | 100.0 | 100.0 |
| VGG16 | 71.6 | 73.7 | 96.8 | 98.7 | 99.4 |

| Model | Clean Acc | AutoAttack attack radius | | | |
|---|---|---|---|---|---|
| | | 0.001 | 0.003 | 0.005 | 0.01 |
| ViT-S/16 | 77.6 | **51.9** | 94.0 | 99.5 | 100.0 |
| ViT-B/16 | 75.7 | 60.2 | 94.6 | 99.4 | 100.0 |
| ViT-L/16 | 79.2 | 53.4 | **91.5** | **99.0** | 100.0 |
| ViT-B/16-Res | 84.0 | 72.3 | 99.1 | 100.0 | 100.0 |
| T2T-ViT-14 | 80.1 | 87.1 | 99.9 | 100.0 | 100.0 |
| T2T-ViT-24 | 82.2 | 79.2 | 99.7 | 100.0 | 100.0 |
| Dist-Deit-S/16 | 79.3 | 56.9 | 96.3 | 99.8 | 100.0 |
| Dist-Deit-B/16 | 81.8 | 57.3 | 96.6 | 99.8 | 100.0 |
| Swin-S/4 | 81.8 | 92.1 | 99.9 | 100.0 | 100.0 |
| SEResNet50 | 75.7 | 78.4 | 99.4 | 100.0 | 100.0 |
| ResNeXt-32x4d-ssl | 80.3 | 93.5 | 100.0 | 100.0 | 100.0 |
| ResNet50-swsl | 81.2 | 91.9 | 100.0 | 100.0 | 100.0 |
| ResNet18 | 70.0 | 85.7 | 99.6 | 100.0 | 100.0 |
| ResNet50-32x4d | 77.6 | 86.8 | 99.8 | 100.0 | 100.0 |
| ShuffleNet | 69.4 | 93.9 | 100.0 | 100.0 | 100.0 |
| MobileNet | 71.9 | 92.2 | 100.0 | 100.0 | 100.0 |
| VGG16 | 71.6 | 83.3 | 99.5 | 100.0 | 100.0 |

### 5.2 ROBUSTNESS UNDER TRANSFER ATTACK

We also conduct transfer attack to test the adversarial robustness in the black-box setting as described in Section 4. We consider attacks with $\ell_\infty$-norm perturbation no larger than 0.1 and present the results in Figure 2. When the ViTs serve as the target models and CNNs serve as the source models, as shown in the lower left of each subplot, the ASR of the transfer attack is quite low. On the other hand, when the ViTs are the source models, the adversarial examples they generate have higher ASR when transferred to other target models. As a result, the first three rows and the last seven columns

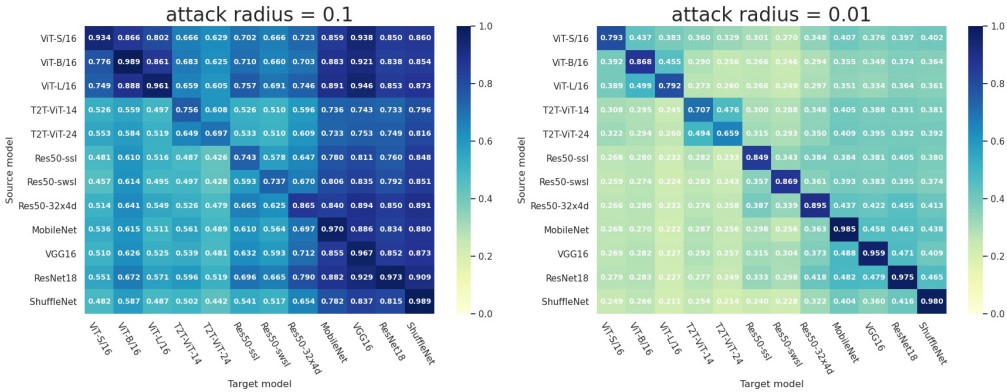

Figure 2: ASR of transfer attack using FGSM with different attack radii. The rows stand for the surrogate models used to generated adversarial examples. The columns stand for the target models. Darker rows correlate to the source models that generate more transferable adversarial examples. Darker columns mean that the target models are more vulnerable to transfer attack. "Res50-ssl" and "Res50-swsl" are in short of "ResNeXt-32x4d-ssl" and "ResNet50-swsl" respectively. Results for more radii can be found in Appendix C.

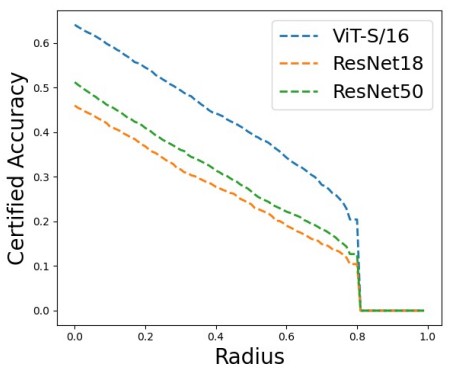

Figure 3: Certification of ViT-S/16, ResNet50 and ResNet18 ImageNet classifiers using denoised randomized smoothing ($\sigma = 0.25$).

Table 3: Results of adversarial training for different models using PGD-7 (7-step PGD attack) and TRADES respectively on CIFAR-10. ViT-B/4 is a variant of ViT-B/16 where we downsample the patch embedding kernel from $16 \times 16$ to $4 \times 4$ to accommodate the smaller image size on CIFAR-10. We report the clean accuracy (%) and robust accuracy (%) evaluated with PGD-10 and AutoAttack respectively. Each model is trained using only 20 epochs to reduce the cost.

| Model | Method | Clean | PGD-10 | AutoAttack |
|---|---|---|---|---|
| PreActResNet18 | PGD-7 | 77.3 | 48.9 | 44.4 |
| | TRADES | 77.6 | 49.4 | 44.9 |
| WideResNet-34-10 | PGD-7 | 80.3 | 52.2 | 48.4 |
| | TRADES | 81.6 | 53.4 | 49.3 |
| ViT-B/4 | PGD-7 | 85.9 | 51.7 | 47.6 |
| | TRADES | 85.0 | 53.9 | 49.2 |

are darker than the others. Besides, for the diagonal lines in the figure where FGSM actually attacks the models in a white-box setting, we can observe that ViTs are less sensitive to attack with smaller radii compared to CNNs, and T2T modules make ViTs more robust to such one-step attack. In addition, adversarial examples transfer well between models with similar structures. As ViT-S/16, ViT-B/16 and ViT-L/16 have similar structures, the adversarial examples generated by them can transfer well to each other, and it is similar for T2T-ViTs and CNNs respectively.

## 5.3 CERTIFIED ROBUSTNESS

**Settings** We train the denoisers using the stability objective for 25 epochs with a noise level of $\sigma = 0.25$, learning rate of $10^{-5}$ and a batch size of 64. We iterate through the ImageNet dataset to calculate the corresponding radii according to Eq. 6, and report the certified accuracy versus different radii as defined in Salman et al. (2020) in Figure 3.

**Results** As shown in Figure 3, ViT-S/16 has higher certified accuracy than ResNet18, showing better certified robustness of vision transformers over CNNs. We also found that, in the same settings, training a Gaussian denoiser for the ResNet18 is harder than for the ViT-S/16. Accuracy of ViT-S/16 with denoiser at noise of $\sigma = 0.25$ is $64.84\%$ ($4.996\%$ without any denoiser), while accuracy of ResNet50 and ResNet18 with denoiser at the same noise is $47.782\%$ ($5.966\%$ without any denoiser).

## 5.4 Adversarial Training

**Settings**  We also conduct a preliminary experiment on adversarial training for ViT. For this experiment we use CIFAR-10 (Krizhevsky et al. (2009)) with $\epsilon = 8/255$ and the ViT-B/16 model. Since originally this ViT was pre-trained on ImageNet with image size $224 \times 224$ and patch size $16 \times 16$ while image size on CIFAR-10 is $32 \times 32$, we downsample the weights for patch embeddings and resize patches to $4 \times 4$, so that there are still $8 \times 8$ patches and we name the new model as ViT-B/4. Though ViT originally enlarged input images on CIFAR-10 for natural fine-tuning and evaluation, we keep the input size as $32 \times 32$ to make the attack radius comparable. For training, we use PGD-7 (PGD with 7 iterations) (Madry et al. (2017)) and TRADES (Zhang et al. (2019)) methods respectively, with no additional data during adversarial training. We compare ViT with two CNNs, ResNet18 (He et al. (2016)) and WideResNet-34-10 (Zagoruyko & Komodakis (2016)). To save training cost, we train each model for 20 epochs only, although some prior works used around hundreds of epochs (Madry et al. (2017); Pang et al. (2020)) and are very costly for large models. We use a batch size of 128, an initial learning rate of 0.1, an SGD optimizer with momentum 0.9, and the learning rate decays after 15 epochs and 18 epochs respectively with a rate of 0.1. While we use a weight decay of $5 \times 10^{-4}$ for CNNs as suggested by Pang et al. (2020) that $5 \times 10^{-4}$ is better than $2 \times 10^{-4}$, we still use $2 \times 10^{-4}$ for ViT as we find $5 \times 10^{-4}$ causes an underfitting for ViT. We evaluate the models with PGD-10 (PGD with 10 iterations) and AutoAttack respectively.

**Results**  We show the results in Table 3. The ViT model achieves higher robust accuracy compared to ResNet18, and comparable robust accuracy compared to WideResNet-34-10, while ViT achieves much better clean accuracy compared to the other two models. Here ViT does not advance the robust accuracy after adversarial training compared to large CNNs such as WideResNet-34-10. We conjecture that ViT may need larger training data or longer training epochs to further improve its robust training performance, inspired by the fact that on natural training ViT is not able to perform well either without large-scale pre-training. And although T2T-ViT improved the performance of natural training when trained from scratch, our previous results in Table 1 and Table 2 show that the T2T-ViT structure may be inherently less robust. We have also tried Wong et al. (2020) which was proposed to mitigate the overfitting of FGSM to conduct fast adversarial training with FGSM, but we find that it can still cause catastrophic overfitting for ViT such that the test accuracy on PGD attacks remains almost 0. We conjecture that this fast training method may be not suitable for pre-trained models or require further adjustments. Our experiments in this section demonstrate that the adversarial training framework with PGD or TRADES is applicable for transformers on vision tasks, and we provide baseline results and insights for future exploration and improvement.

## 6 Reasoning of Adversarial Robustness

In this section, we present extended analysis to dissect the source of improved adversarial robustness in ViTs. We also verify that ViT's improvement is not caused by insufficient attack optimization, and an explanation from Hopfield network perspective is provided (see Appendix D for details).

**Learning Low-Level Structures Makes Models Less Robust**  One interesting and perhaps surprising finding is that ViTs have worse robustness when modules that help to learn local structures are added ahead of the transformer blocks. For example, T2T-ViT adds several T2T modules to the head of ViT which iteratively aggregates the neighboring tokens into one token in each local perceptive field. ViT-B/16-Res takes the features generated by ResNet as inputs, which has the same effect as incorporating a trained CNN layer in front of the transformer blocks. Both modules help to learn local structures like edges and lines (Yuan et al. (2021)).

When the features learned by ResNet are introduced, the ASR of ViT-B/16 rises from $51.1\%$ to $54.5\%$ of ViT-B/16-Res under PGD attack, and from $60.2\%$ to $72.3\%$ under AutoAttack, with attack radius $\epsilon = 0.001$. A similar phenomenon can be observed by comparing the ASR of ViTs and T2T-ViTs. The ASR of T2T-ViT-14 is $18.3\%$ higher under PGD attack and $35.2\%$ higher under AutoAttack compared with the ASR of ViT-S/16, under attack radius $\epsilon = 0.001$.

One possible explanation is that the introduced modules improve the classification accuracy by remembering the low-level structures that repeatedly appear in the training dataset. These structures such as edges and lines are high-frequent and sensitive to perturbations. Learning such features

makes the model more vulnerable to adversarial attacks. Examination of this hypothesis is conducted in Section 6.1.

**Increasing the Proportion of Transformer Blocks Can Improve Robustness**   Hendrycks et al. (2019) mentioned that larger model does not necessarily imply better robustness. It can be confirmed by our experiments where ViT-S/16 shows better robustness than larger ViT-B/16 under both PGD attack and AutoAttack. In this case, simply adding transformer blocks to the classifier cannot guarantee better robustness. However, we recognize that for mixed architecture that has both T2T and transformer blocks, it is useful to improve adversarial robustness by increasing the proportion of the transformer blocks in the model. As shown in Tables 1 and 2, T2T-ViT-24 has lower ASR than T2T-ViT-14 under both attacks. Besides the transformer block, we find that other attention mechanism modules such as SE block also improves adversarial robustness – as SEResNet50 has the least proportion of attention, the ASR of SEResNet50 is higher than ViT and T2T-ViT models but lower than other pure CNNs. These two findings are coherent since the attention mechanism is fundamental in transformer blocks.

**Pre-training without Adversarial Training Does Not Improve Robustness**   Pre-training is critical for ViTs to achieve competitive standard accuracy with CNNs trained from scratch (Dosovitskiy et al. (2020)). However, pre-training may not be the main reason of better robustness. To illustrate this point, we include CNNs pre-trained on large datasets and fine-tuned on ImageNet-1k to check the effect of pre-training on adversarial robustness. CNNs pre-trained on large datasets IG-1B-Targeted (Mahajan et al. (2018)) and YFCC100M (Thomee et al. (2015)) that are even larger than ImageNet-21k used by ViT, ResNet50-swsl and ResNeXt-32x4d-ssl, still have similar or even higher ASR than ResNet18 and ResNet50-32x4d that are not pre-trained. This supports our observation that pre-training in its current form may not be able to improve adversarial robustness. Hendrycks et al. (2019) also reported that pre-training without adversarial training techniques cannot improve adversarial robustness. The resilience of ViT against perturbations corresponds more to the transformer structure rather than pre-training.

## 6.1 FREQUENCY STUDY AND FEATURE VISUALIZATION

Table 4: Frequency study and the ASR (%) of the target models against PGD attack. In the "Low-pass" column, only low-frequent adversarial perturbations are preserved and added to the input images. In the "High-pass" column, only high-frequent perturbations can pass through the filter. The "Full-pass" mode is the same as the traditional PGD attack. We set the attack step fixed to 40 and vary the attack radius to different values as shown in the second row.

| | Low-pass | | | | | High-pass | | | | | Full-pass | | | | |
|---|---|---|---|---|---|---|---|---|---|---|---|---|---|---|---|
| Model | 0.001 | 0.003 | 0.005 | 0.01 | 0.1 | 0.001 | 0.003 | 0.005 | 0.01 | 0.1 | 0.001 | 0.003 | 0.005 | 0.01 | 0.1 |
| ViT-S/16 | 26.0 | 31.9 | 35.3 | 40.2 | 43.8 | 29.2 | 39.3 | 49.4 | 59.6 | 76.6 | 44.6 | 75.4 | 89.8 | 99.0 | 100.0 |
| ViT-B/16 | 28.1 | 35.7 | 39.7 | 44.2 | 50.4 | 33.7 | 46.9 | 56.0 | 66.6 | 78.1 | 51.1 | 85.4 | 94.0 | 99.1 | 100.0 |
| ViT-L/16 | 25.1 | 35.9 | 41.7 | 49.8 | 58.0 | 27.1 | 37.7 | 43.4 | 52.5 | 71.1 | 44.9 | 76.6 | 90.1 | 98.2 | 100.0 |
| ViT-B/16-Res | 16.9 | 18.6 | 19.6 | 21.0 | 24.9 | 37.1 | 70.8 | 84.0 | 92.7 | 96.7 | 54.5 | 91.6 | 97.7 | 99.9 | 100.0 |
| T2T-ViT-14 | 22.0 | 22.8 | 24.0 | 24.2 | 25.7 | 50.4 | 79.5 | 90.9 | 96.9 | 98.6 | 62.9 | 93.0 | 98.2 | 100.0 | 100.0 |
| T2T-ViT-24 | 19.8 | 20.8 | 21.6 | 22.3 | 25.6 | 41.7 | 68.9 | 82.3 | 91.8 | 96.9 | 52.3 | 87.7 | 96.6 | 99.8 | 100.0 |
| ResNet50-swsl | 21.8 | 25.1 | 26.3 | 28.4 | 27.5 | 54.7 | 87.6 | 95.0 | 97.8 | 96.5 | 75.3 | 97.1 | 98.6 | 99.6 | 100.0 |
| ResNet50-32x4d | 25.0 | 33.7 | 37.3 | 41.0 | 38.5 | 52.3 | 82.9 | 92.6 | 96.7 | 96.5 | 71.8 | 96.8 | 98.8 | 99.6 | 99.9 |

Here we conduct a frequency study and feature visualization to support our claim that ViTs are more robust because they learn to focus less on high-frequency features than other models. Such property can be diminished when introducing other modules to the ViT's structure, causing the inferior adversarial robustness of hybrid ViTs.

**Frequency Study**   We design a frequency study to verify our hypothesis that ViTs are adversarially more robust compared with CNNs because ViTs learn less high-frequency features. As defined in equation 3, for adversarial perturbations generated by PGD attack, we first project them to the frequency domain by DCT. We design three frequency filters as shown in Figure 5 in Appendix B: the full-pass filter, the low-pass filter, and the high-pass filter. We take $32 \times 32$ pixels in the low-frequency area out of $224 \times 224$ pixels as the low-pass filter, and $192 \times 192$ pixels in the high-frequency area as the high-pass filter. Each filter allows only the corresponding frequencies to pass through – when the adversarial perturbations go through the low-pass filter, the high-frequency

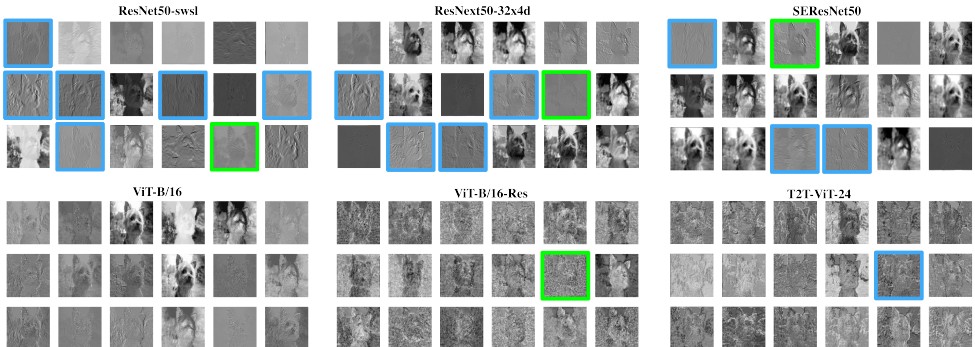

Figure 4: Feature visualization: The learned low-level structure features are highlighted in blue (obviously perceptible) and green (minorly perceptible). The CNNs in the first row learn more low-level features compared with the ViTs in the second row. The ViTs pay more attention to the low-level structures and their feature maps become noisier when ResNet features are introduced (ViT-B/16-Res) or neighboring tokens are aggregated into one token recursively (T2T-ViT-24).

components are filtered out and vice versa, and the full-pass filter makes no change. We then apply these filters to the frequencies of the perturbations, and project them back to the spacial domain with the IDCT. We test the ASR under different frequency areas, and show the results in Table 4.

The ASR of ViT is relatively low in the "High-pass" column when only the high-frequencies of the perturbations are preserved. In contrast, CNNs show significantly higher ASR in the "High-pass" column than in the "Low-pass" column. It reflects that CNNs tend to be more sensitive to high-frequency adversarial perturbations compared to ViTs. We also observe that adding modules that learn low-level structures makes the models more sensitive to high-frequency perturbations. T2T-ViT-14, T2T-ViT-24 and ViT-B/16-Res have higher ASR in the "High-pass" column and lower ASR in the "Low-pass" column compared with vanilla ViTs, which verifies our hypothesis that low-level features are less adversarially robust. Besides, when adding more transformer blocks to the T2T-ViT model, the model becomes less sensitive to the high frequencies of the adversarial perturbations, e.g., the T2T-ViT-24 has an 8.7% lower ASR than that of the T2T-ViT-14 in the "High-pass" column.

**Feature Visualization** We follow the work of Yuan et al. (2021) to visualize the learned features from the first blocks of the target models in Figure 4. We resize the input images to a resolution of $224 \times 224$ for CNNs and a resolution of $1792 \times 1792$ for ViTs and T2T-ViTs such that the feature maps from the first block are in the same shape of $112 \times 112$. Low-level features like lines and edges are highlighted in blue (obviously perceptible) and green (minorly perceptible). As shown in Figure 4, CNNs like ResNet50-swsl and ResNet50-32x4d learn features with obvious edges and lines. Minorly perceptible low-level features are learned by T2T-ViT-24 and ViT-B/16-Res. While it is hard to observe such information in the features maps learned by ViT-B/16.

The feature visualization combined with the frequency study shows that the model's vulnerability under adversarial perturbations is highly relative to the model's tendency to learn low-level high-frequency features. Techniques that help the model learn such features may improve the performance on clean data but at the risk of sacrificing adversarial robustness.

## 7    CONCLUSION

This paper presents the first study on the robustness of ViTs against adversarial perturbations. Our results indicate that ViTs are more robust than CNNs on the considered adversarial attack and certified robustness settings. Moreover, we show that the features learned by ViTs contain less low-level information, contributing to improved robustness against adversarial perturbations that often contain high-frequency components; introducing convolutional blocks in ViTs can facilitate learning low-level features but has a negative effect on adversarial robustness and makes the models more sensitive to high-frequency perturbations. We also demonstrate adversarial training for ViT. Our work provides a deep understanding of the intrinsic robustness of ViTs and can be used to inform the design of robust vision models based on the transformer structure.

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

## SUPPLEMENTAL MATERIAL

In this supplemental material, we provide more analysis and results in our experiments.

## A   TARGET MODELS

Summary of the target models investigated in the main text are shown in Table 5. The weights of these models are all publicly available at Paszke et al. (2019); Wightman (2019) such that our experiments can be easily reproduced.

Table 5: Comparison of the target models investigated in the main text.

| Model | ViT backbone | | | | | Pretraining | |
| | Layers | Hidden size | Attention | Params | Pretraining dataset | Scale |
| --- | --- | --- | --- | --- | --- | --- |
| ViT-S/16 | 8 | 786 | Self-attention | 49M | ImageNet-21K | 14M |
| ViT-B/16 | 12 | 786 | Self-attention | 87M | ImageNet-21K | 14M |
| ViT-L/16 | 24 | 1024 | Self-attention | 304M | ImageNet-21K | 14M |
| ViT-B/16-Res | 12 | 786 | Self-attention | 87M | ImageNet-21K | 14M |
| T2T-ViT-14 | 14 | 384 | Self-attention | 22M | - | - |
| T2T-ViT-24 | 24 | 512 | Self-attention | 64M | - | - |
| DeiT-S/16 | 12 | 384 | Self-attention | 22M | - | - |
| Dist-DeiT-B/16 | 12 | 768 | Self-attention | 87M | - | - |
| Swin-S/4 | (2,2,18,2) | 96 | Self-attention | 50M | - | - |
| SEResNet50 | - | - | Squeeze-and-Excitation | 28M | - | - |
| ResNeXt-32x4d-ssl | - | - | - | 25M | YFCC100M | 100M |
| ResNet50-swsl | - | - | - | 26M | IG-1B-Targeted | 940M |
| ResNet18 | - | - | - | 12M | - | - |
| ResNet50-32x4d | - | - | - | 25M | - | - |
| ShuffleNet | - | - | - | 2M | - | - |
| MobileNet | - | - | - | 4M | - | - |
| VGG16 | - | - | - | 138M | - | - |

## B   FREQUENCY FILTERS

We show the design of frequency-filters in Figure 5.

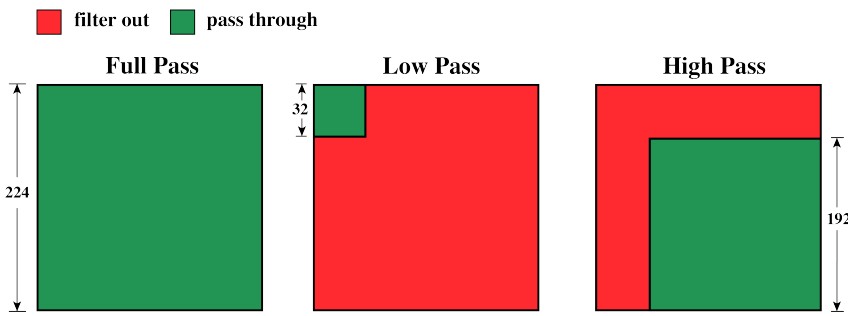

Figure 5: Filters for the frequency-based attack. The frequencies corresponding to the red part are filtered out, and the frequencies corresponding to the green part can pass through. "Full Pass" means all of the frequencies are preserved. "Low Pass" means only low-frequent components are preserved. "High Pass" preserves the high-frequent part.

## C   TRANSFER ATTACK RESULTS

Transfer attack results using more attack radii are provided in Figure 6

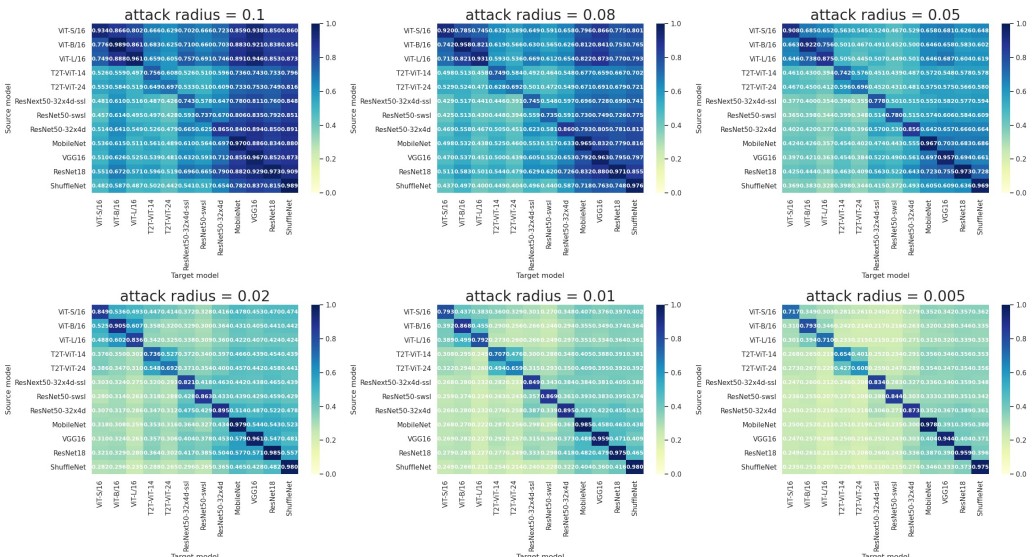

Figure 6: ASR of transfer attack using FGSM with different attack radii. The rows stand for the surrogate models used to generated adversarial examples in the white-box attack approach. The columns stand for the target models. Darker rows correlate to the source models that generate more transferable adversarial examples. While darker columns mean that the target models are more vulnerable to the transfer attack."Res50-ssl" and "Res50-swsl" are in short of "ResNeXt-32x4d-ssl" and "ResNet50-swsl" respectively.

## D  THE SOURCE OF ADVERSARIAL ROBUSTNESS

In this section we examine the source of the adversarial robustness revealed in our experiments.

**The improved robustness of ViT is not caused by insufficient attack optimization.** We first demonstrate that the better robustness of ViTs in white-box attacks is not caused by the difficult optimization in ViT by plotting the loss landscape with sufficient attack steps.

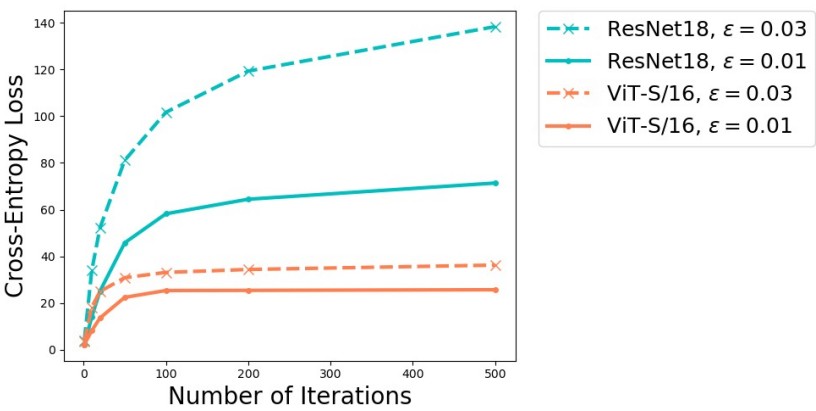

Figure 7: Cross entropy loss versus varying PGD attack steps for ViT-S/16 and RestNet18. The dashed lines corresponds to larger attach radius of 0.03 and the full lines to smaller attack radius of 0.01.

Figure 7 shows the cross entropy loss versus varing PGD attack steps for ViT-S/16 and ResNet18. As shown in the figure, ViT's loss curves converge at a much lower value than RestNet18, suggesting that the improved robustness of ViT is not caused by insufficient attack optimization.

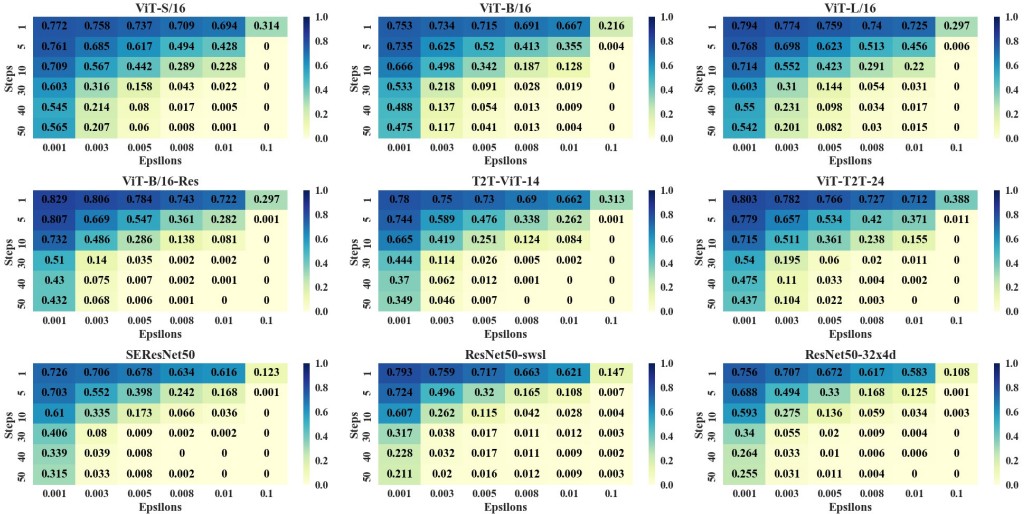

Figure 8: Adversarial accuracy of the target models against PGD attack with different attack radii ("eps") and attack steps ("steps"). When the attack radius and attack steps are increased, the adversarial accuracy of the target model decreases to zero. Darker blocks stand for more robust models against PGD attack.

Figure 8 shows the robust accuracy of more target models against PGD attack with different attack radii ("eps") and attack steps ("steps"). Vision transformers have darker blocks than CNNs', which stands for their superior adversarial robustness against PGD attack.

**A Hopfield Network Perspective** The equivalence between the attention mechanism in transformers to the modern Hopfield network (Krotov & Hopfield (2016)) was recently shown in Ramsauer et al. (2020). Furthermore, on simple Hopfield network (one layer of attention-like network) and dataset (MNIST), improved adversarial robustness was shown in Krotov & Hopfield (2018). Therefore, the connection of attention in transformers to the Hopfield network can be used to explain the improved adversarial robustness for ViTs.

# E  EXPERIMENTS ON CIFAR-10

We choose the ImageNet as the benchmark because ViTs can hardly converge when training directly on small datasets like Cifar. Therefore, we finetune the ViTs instead. As shown in Table 6, ViT-B/4 performs higher robust accuracy than WideResNet, which is consistent with the trend on ImageNet.

Table 6: Robust accuracy of ViT-B/4 and WideResNet against PGD-10 attack with different attack radii.

| Model | 0.001 | 0.003 | 0.01 | 0.03 |
|---|---|---|---|---|
| ViT-B/4 | 0.9202 | 0.6242 | 0.0994 | 0.0103 |
| WideResNet | 0.7744 | 0.5923 | 0.0854 | 0.0000 |

# F  EXPERIMENTS ON SOTA VIT STRUCTURES

In this section, we supplement the experimental results of recently proposed SOTA ViTs.

**Swin-Trasnformer (Liu et al., 2021)**    computes the representations with shifted windows scheme which brings greater efficiency by limiting self-attention computation to non-overlapping local windows while also allowing for cross-window connection.

**DeiT (Touvron et al., 2021)**    further improves the ViTs' performance using data augmentation or distillation from CNN teachers with an additional distillation token.

**SAM-ViT (Chen et al., 2021)**    uses sharpness-aware minimization (Foret et al., 2020) to train ViTs from scratch on ImageNet without large-scale pretraining or strong data augmentations.

Table 7 summarizes the information of models investigated in our experiments. The window size of the swin transformers in Table 7 is 7. The pre-trained weights of these models are available in `timm` package.

Table 7: SOTA ViT models investigated in our experiments.

| Model | Layers | Hidden size | Heads | Params |
|-------|--------|-------------|-------|--------|
| Deit-T/16 (Touvron et al., 2021) | 12 | 192 | 3 | 6M |
| Deit-S/16 (Touvron et al., 2021) | 12 | 384 | 6 | 22M |
| Deit-B/16 (Touvron et al., 2021) | 12 | 768 | 12 | 87M |
| Dist-Deit-T/16 (Touvron et al., 2021) | 12 | 192 | 3 | 6M |
| Dist-Deit-S/16 (Touvron et al., 2021) | 12 | 384 | 6 | 22M |
| Dist-Deit-B/16 (Touvron et al., 2021) | 12 | 768 | 12 | 87M |
| ViT-SAM-B/16 (Chen et al., 2021) | 12 | 768 | 12 | 87M |
| ViT-SAM-B/32 (Chen et al., 2021) | 12 | 768 | 12 | 88M |
| Swin-T/4 (Liu et al., 2021) | (2,2,6,2) | 96 | (3,6,12,24) | 28M |
| Swin-S/4 (Liu et al., 2021) | (2,2,18,2) | 96 | (3,6,12,24) | 50M |
| Swin-B/4 (Liu et al., 2021) | (2,2,18,2) | 128 | (4,8,16,32) | 88M |
| Swin-L/4 (Liu et al., 2021) | (2,2,18,2) | 192 | (6,12,24,48) | 197M |

Table 8: Robust accuracy (%) of ViTs described in Table 7 against 40-step PGD attack with different attack radii, and also the clean accuracy ("Clean"). A model is considered to be more robust if the robust accuracy is higher.

| Model | Clean | 0.001 | 0.003 | 0.005 | 0.01 |
|-------|-------|-------|-------|-------|------|
| Deit-T/16 | 72.3 | 36.8 | 8.3 | 2.6 | 0.3 |
| Deit-S/16 | 77.7 | 48.9 | 17.6 | 7.1 | 1.1 |
| Deit-B/16 | 81.3 | 46.6 | 14.3 | 6.0 | 0.9 |
| Dist-Deit-T/16 | 74.4 | 40.6 | 5.7 | 0.7 | 0.2 |
| Dist-Deit-S/16 | 79.3 | 52.4 | 15.1 | 4.3 | 0.3 |
| Dist-Deit-B/16 | 81.8 | 55.6 | 17.7 | 4.5 | 0.4 |
| ViT-SAM-B/16 | 76.7 | 63.4 | 37.0 | 20.1 | 3.8 |
| ViT-SAM-B/32 | 63.8 | 53.2 | 32.3 | 19.7 | 3.1 |
| Swin-T/4 | 78.8 | 33.5 | 6.0 | 1.2 | 0.1 |
| Swin-S/4 | 81.8 | 40.0 | 12.4 | 3.2 | 0.2 |
| Swin-B/4 | 82.3 | 38.8 | 11.1 | 4.1 | 0.3 |
| Swin-L/4 | 84.2 | 38.7 | 11.1 | 2.9 | 0.4 |

Table 8 shows the clean and robust accuracy of ViTs in Table 7 against 40-step PGD attack with different radii. And results for AutoAttack are shown in Table 9. Swin-transformers introduce shifted windows scheme that limit self-attention computation to non-overlapping local windows, which harms the robustness as Tokens-to-Token scheme according to the above results.

Table 9: Robust accuracy (%) of ViTs described in Table 7 against AutoAttack with different attack radii, and also the clean accuracy ("Clean"). A model is considered to be more robust if the robust accuracy is higher.

| Model | Clean | 0.001 | 0.003 | 0.005 | 0.01 |
|---|---|---|---|---|---|
| Deit-T/16 | 72.3 | 23.4 | 0.5 | 0.0 | 0.0 |
| Deit-S/16 | 77.7 | 30.2 | 1.2 | 0.0 | 0.0 |
| Deit-B/16 | 81.3 | 20.4 | 0.3 | 0.1 | 0.0 |
| Dist-Deit-T/16 | 74.4 | 31.1 | 0.8 | 0.1 | 0.0 |
| Dist-Deit-S/16 | 79.3 | 43.1 | 3.7 | 0.2 | 0.0 |
| Dist-Deit-B/16 | 81.8 | 42.7 | 3.4 | 0.2 | 0.0 |
| ViT-SAM-B/16 | 76.7 | 59.8 | 26.0 | 8.4 | 0.1 |
| ViT-SAM-B/32 | 63.8 | 48.9 | 23.6 | 9.7 | 0.8 |
| Swin-T/4 | 78.8 | 6.8 | 0.1 | 0.0 | 0.0 |
| Swin-S/4 | 81.8 | 7.9 | 0.1 | 0.0 | 0.0 |
| Swin-B/4 | 82.3 | 2.4 | 0.1 | 0.0 | 0.0 |
| Swin-L/4 | 84.2 | 4.3 | 0.1 | 0.0 | 0.0 |

## G  DISCUSSION OF CONCURRENT AND RELATED WORKS

In the context of computer vision, one earliest relevant work is Alamri et al. (2020) which applies transformer encoder in the object detection task and reports better adversarial robustness. But the model they considered is a mix of CNN and transformer instead of the ViT model considered in this paper. Besides, the attacks they applied were relatively weak, and there lacks study and explanation on the benefit of adversarial robustness brought by the transformers.

Recently, there are many concurrent or follow-up works investigating the adversarial robustness of vision transformers. Many of them have cited our work and shown valuable discussion or extended experiments. We acknowledge their contributions below. Mahmood et al. (2021) test the adversarial robustness of the transformer in both white-box and black-box settings, analyze the security of a simple ensemble defense of CNNs and transformers, and find such ensemble defense is effective for the black-box setting. Qin et al. (2021) and Salman et al. (2021) investigate the adversarial robustness of ViTs through the lens of their special patch-based architectural structure. Naseer et al. (2021b) investigate whether the weak transferability of adversarial patterns from high-performing ViT models, as reported in our work, is a result of weak features or a weak attack. Aldahdooh et al. (2021); Naseer et al. (2021a); Paul & Chen (2021); Tang et al. (2021) study the adversarial robustness from different views, e.g., preprocessing defense methods (Aldahdooh et al., 2021), shape recognition capability (Naseer et al., 2021a), and natural adversarial examples (Paul & Chen, 2021; Tang et al., 2021).

Our work differs from these concurrent and related works by focusing more on the origin of the adversarial robustness of vision transformers. We discuss the superior adversarial robustness of ViTs through the lens of frequency, and find that the ViTs are especially robust to high-frequency adversarial perturbations. We also apply denoised randomized smoothing and show ViT also has superior certified robustness than CNN models. Our study provides insight on understanding the source of ViT's adversarial robustness and designing more robust architectures.

## H  ROBUSTNESS AGAINST ADVERSARIAL DEFORMATION

Besides additively perturbing the correctly classified image, ADef (Alaifari et al., 2018) iteratively applies small deformations to the clean data. We show the robust accuracy against such perturbations in Table 10, which is in accordance to the results of PGD and AutoAttack.

Table 10: Robust accuracy (%) against AFef under the default setting described in Alaifari et al. (2018).

| Model | ViT-S/16 | VGG16 | DenseNet | MobileNet | ResNet18 |
|---|---|---|---|---|---|
| **Robust Accuracy** | **12.4** | 10.8 | 11.1 | 11.7 | 11.8 |

