# OpenReview forum: "On the Adversarial Robustness of Vision Transformers"
_ICLR.cc/2022/Conference — ICLR 2022 Submitted_

### Official Review · Reviewer_WHoT · 2021-10-30

**Correctness:** 3
**Technical Novelty And Significance:** 3
**Empirical Novelty And Significance:** 3
**Recommendation:** 5
**Confidence:** 4

**Main Review:**

Strengths:
The study on the adversarial robustness of ViT is comprehensive.
The analyst behind the observation is provided.
The paper is clearly written and easy to follow.

Weakness:
1. The authors state that ViT is more robust than CNN. It is only true when a weak attack is applied. When a standard attack (with perturbation range epsilon=0.03/0.01), both ViT and CNN can be fooled with 100% fooling rate. The observation indicates that both are equally vulnerable under standard attack. Hence, the statements or claims made in this paper should be carefully formulated.

2. The observations on the difference in model robustness are presented in this work. However, the difference is always attributed to the model architecture difference in this work. Compared to ResNet, the ViT and DeiT are trained/finetuned in a different setting, e.g. different data augmentation, different training schedules. It is not clear how the different training/finetuning settings contribute to the different model robustness. It might be too early to attribute the observation to model architectures.

3. Similarly, the author lists the following point as one of their findings:
Pre-training on larger datasets does not improve adversarial robustness though it is critical for training ViT.
However, previous work shows pre-training can improve model robustness [1]. More discussion should be provided hier. The claims should be careful.
4. In Figure 3, the author shows ViT-small shows higher certified robustness than ResNet18. In the work DeiT, they show that the CNN counterpart of DesT-small is ResNet50 instead of ResNet18. If it is too hard to compare them in a 100% fair fashion. Please show the curves of both ResNet50 and ResNet18.

5. The adversarial training on ViT is also studied in this work. However, the experiments are conducted on CIFAR10 datasets. The ViT is expected to behave well only on a large dataset. What is the performance of ViT in a standard-setting on CIFAR10 dataset?

[1] Hendrycks, Dan, Kimin Lee, and Mantas Mazeika. "Using pre-training can improve model robustness and uncertainty." International Conference on Machine Learning. PMLR, 2019.

**Summary Of The Paper:**

This work studies the adversarial robustness of vision transformers comprehensively. The author first presents their finding via empirical experiments. Concretely, they show the robustness of vision transformers from the perspective of adversarial attack, transferability, and certified robustness, adversarial training defense. Then, they analyze the reason behind the observation.

**Summary Of The Review:**

ViT demonstrates the potential to work as an alternative to CNNs. The adversarial robustness of ViT is indeed an important topic. As claimed by the author, this work makes the first comprehensive study on this topic. However, some concerns listed in weaknesses above remain to be addressed. Therefore, I rate this paper blew the acceptance threshold. I happy to raise my rating if the concerns are well addressed.

---

> ### Author Response · Authors · 2021-11-21
> **Response to Reviewer WHoT (1)**
>
> >Q1:\
> The authors state that ViT is more robust than CNN. It is only true when a weak attack is applied. When a standard attack (with perturbation range epsilon=0.03/0.01), both ViT and CNN can be fooled with 100% fooling rate. The observation indicates that both are equally vulnerable under standard attack. Hence, the statements or claims made in this paper should be carefully formulated.
>
> A1:\
> Thanks for your valuable comments. When we are making robustness arguments we indeed assume a given perturbation budget that is the same for both types of neural networks. We agree that our claimed robustness holds for small attack radii but their margin becomes smaller as one increases the perturbation budget. However, we believe our findings and analysis of improved robustness for ViTs (for small perturbation) still give valuable insights for understanding contributions from the architecture-level. Also, we believe our result of showing ViT has higher certified robustness than CNN is new to this field (Sec. 5.3). Finally, one can apply adversarial training, as discussed in Sec. 5.4, to further strengthen the adversarial robustness of ViTs against larger perturbations.
>
> >Q2:\
> The observations on the difference in model robustness are presented in this work. However, the difference is always attributed to the model architecture difference in this work. Compared to ResNet, the ViT and DeiT are trained/finetuned in a different setting, e.g. different data augmentation, different training schedules. It is not clear how the different training/finetuning settings contribute to the different model robustness. It might be too early to attribute the observation to model architectures.
>
> A2:\
> We agree that training settings, especially the data augmentation, may raise a problem for fair comparison. However, we believe it is also important to compare adversarial robustness of best available pre-trained models from ViTs and CNNs, when they are separately trained to “optimum” that favors their own training practices.  Trying to address this concern from a different angle, we provided a certified robustness analysis using denoised randomized smoothing (Sec. 5.3), which applies to any model trained in different training settings by training a denoiser for each model before conducting certified robustness evaluation, and thus provides a relatively fair comparison of certified robustness. The results suggest that ViT has better certified robustness which could be attributed to the model design.
>
> >Q3:\
> Similarly, the author lists the following point as one of their findings: Pre-training on larger datasets does not improve adversarial robustness though it is critical for training ViT. However, previous work shows pre-training can improve model robustness [1]. More discussion should be provided hier. The claims should be careful.
>
> A3:\
> We referred to [1] in Section 6 and talked about the effect brought by pre-training. We reached the same conclusion as [1], that, in our original words, “ Hendrycks et al. (2019) also reported that pre-training without adversarial training techniques cannot improve adversarial robustness. The resilience of ViT against perturbations corresponds more to the transformer structure rather than pre-training.” We by default regard the pre-training is done in a standard setting without adversarial training. But considering it may raise confusion, we have revised the conclusion by adding “without adversarial training” as a more clear expression.
>
> >Q4:\
> In Figure 3, the author shows ViT-small shows higher certified robustness than ResNet18. In the work DeiT, they show that the CNN counterpart of DesT-small is ResNet50 instead of ResNet18. If it is too hard to compare them in a 100% fair fashion. Please show the curves of both ResNet50 and ResNet18.
>
> A4:\
> In the updated Figure 3, we show the certified accuracy of both ResNet50 and ResNet18 compared with ViT-S/16. Clearly, ViT-S/16 attains the best certified accuracy.
>
> [1] "Using pre-training can improve model robustness and uncertainty.".

---

> ### Author Response · Authors · 2021-11-21
> **Response to Reviewer WHoT (2)**
>
> >Q5:\
> The adversarial training on ViT is also studied in this work. However, the experiments are conducted on CIFAR10 datasets. The ViT is expected to behave well only on a large dataset. What is the performance of ViT in a standard-setting on CIFAR10 dataset?
>
> A5:\
> About CIFAR-10 experiments for adversarial training: We emphasize that the ViT used for adversarial training is **based on a pre-trained model**. We start from a pre-trained model, downsample the patch embeddings, and then do adversarial fine-tuning. When ViT is pre-trained, it can behave well on a small dataset. The original ViT paper (Dosovitskiy et al., 2020) also has fine-tuning experiments on CIFAR-10 with clean accuracy above 99%. For the model we used with downsampled patch embeddings, on CIFAR-10, we train it with standard training for 20 epochs, and we get a clean accuracy of 98.36%. Even if we only fine-tune for 1 epoch, the clean accuracy is also already 96.31%.

---

> > ### Comment · Reviewer_WHoT · 2021-11-25
> > **Response to A5:**
> >
> > In this experiment, the author compares ResNet18 and ViT under the adversarial training setting. Is the applied ResNet18 also pre-trained on the same setting as in ViT? As stated in [1], ResNet18 matches ViT-tiny in terms of the number of parameters, while ViT matches ResNet50. What is the conclusion we get from the adversarial training experiments? Is the conclusion reliable?
> >
> > [1] Touvron, H., Cord, M., Douze, M., Massa, F., Sablayrolles, A., & Jégou, H. (2021, July). Training data-efficient image transformers & distillation through attention. In International Conference on Machine Learning (pp. 10347-10357). PMLR.

---

> > > ### Author Response · Authors · 2021-11-26
> > > **Response**
> > >
> > > In our experiment setting ResNet18 is not pre-trained. Although ViT is pre-trained, it is still not pre-trained on large-scale data with adversarial training, so it may be a reason that ViT does not advance the robust accuracy compared to CNN after adversarial training.
> > >
> > > We would like to remark that this part (Section 5.4) is a preliminary study about the feasibility of applying adversarial training to ViT. As stated in the paper, we aim to provide baseline results of adversarial training to position ViT’s performance and gain insights for future exploration and improvement.  The conclusion is not about whether adversarial training on ViT is better than that on CNN or not, but rather on “how well will the adversarial training techniques in CNN perform when applied to ViT?” Meanwhile, we identify methods that were shown to work well for CNN but do not work well for ViT yet: 1) Wong et al., 2020’s fast FGSM, 2) and the weight decay factor studied by Pang et al., 2020.

---

### Official Review · Reviewer_x14o · 2021-11-01

**Correctness:** 3
**Technical Novelty And Significance:** 2
**Empirical Novelty And Significance:** 3
**Recommendation:** 5
**Confidence:** 3

**Main Review:**

1. Some findings lack explanations and reasons. For example, it is important to give a reason why ViTs have better adversarial robustness than convolutional neural networks. Does this advantage come from the splitted several tokens in ViTs? If yes, could you please conduct more experiments on different sizes of tokens? It would be better to provide more explanations for this.

2. More general Transformer architectures should be explored in the paper. The authors mainly focus on ViTs, and the observations are mainly for ViTs. Recently, Swin Transformer gains more attention in computer vision tasks. The authors only discuss one type of Swin Transformer, i.e., Swin-S/4. It would be better to explore more types of Transformer and draw conclusions for the more general cases of Transformer.

3. The experiments should be improved.  The authors only consider PGD attack and AutoAttack. These two types of attack methods are linear, i.e., add small perturbation on clean data. It would be better to conduct more experiments on other nonlinear attack methods, e.g., ADef [1], which applies small deformations to the clean data.

    [1] ADef : an Iterative Algorithm to Construct Adversarial Deformations. ICLR 2019

4. In Table 1, the attack success rate (ASR) of  Deit-S/16 is the best result, but the ASR of ViT-L/16 is 98.2. Is there a mistake in these two results?


**Summary Of The Paper:**

This paper provides a comprehensive study on the robustness of ViTs against adversarial perturbations. The authors found that 1) ViTs has better adversarial robustness than convolutional neural networks; 2) Introducing convolutional or tokens-to-token blocks can improve the classification accuracy but at the cost of the adversarial robustness; 3) More proportion of transformers has better robustness; 4) Pre-training on larger datasets does not improve adversarial robustness; 5) Adversarial training is applicable to ViTs. In addition, many experiments verify the findings on white-box, transfer attack settings and adversarial training.

**Summary Of The Review:**

It would be better to provide more explanations for some findings, and further improve the experiments for more general Transformer cases and different types of attack methods.

---

> ### Author Response · Authors · 2021-11-21
> **Response to Reviewer x14o (1)**
>
> >Q1:\
> Some findings lack explanations and reasons. For example, it is important to give a reason why ViTs have better adversarial robustness than convolutional neural networks. Does this advantage come from the splitted several tokens in ViTs? If yes, could you please conduct more experiments on different sizes of tokens? It would be better to provide more explanations for this.
>
> A1:\
> In our study, we have provided several explanations and design experiments to explain the improved robustness of ViTs, mostly in Sec. 6 and other parts of this paper. We specify our findings below.
>
> **[Frequency study and feature visualization (Sec. 6.1)]** In our feature visualization experiments shown in Figure 4, features learned by CNNs show obvious lines and edges which correspond to the high-frequency/low-level information in the images. While features learned by ViTs contain less such information. Therefore, it is natural for us to investigate the behavior of ViT against the high/low-frequency adversarial perturbations. In section 6.1, we show ViT is more robust than CNN especially against the high-frequent perturbations (i.e. adversarial perturbations with low-frequent components filtered out). But the ASR against low-frequent perturbations (i.e. adversarial perturbations with high-frequent components filtered out) is similar for CNNs and ViTs. There are some references that support our work:  [1] notices CNN’s ability in capturing the high-frequency components (HFC) of images, which are almost imperceptible to a human but critical for CNNs to boost accuracy (as shown in their Figure 2). As a result, the model uses such imperceptible HFC to make predictions, leading to generalization behaviors counter-intuitive to humans such as adversarial examples. [2] says their experiments further confirmed the robustness of ViT we observed may be due to the lack of convolutional layers, which leads to a bias towards high-frequency adversarial examples.
>
> **[Certified robustness (Sec. 5.3) ]** We also provide a certified robustness analysis using denoised smoothing, which is to our best knowledge unexploited by other concurrent works. The denoised smoothing applies to any model. The result suggests that ViT has better certified robustness which could be attributed to the model architecture.
>
> **[Portion of transformer blocks in hybrid ViTs (Sec. 6)]** Transformer uses self-attention so it tends to focus more on high-level information, while CNNs use convolution and thus focus more on low-level information. Our hybrid Conv-ViT verifies their effect on robustness, showing ViTs using features processed by convolution blocks are more vulnerable to adversarial attack. Our finding is also consistent with other (later) works, e.g., [2] further supports our conclusion by showing that the lack of convolutional operations in ViTs is responsible for this greater robustness to high-frequency attacks. Based on this finding, the reviewer’s intuition is correct because the split of tokens mentioned by the reviewer is a consequence of using (full) transformer blocks in ViTs.
>
> **[Connection to Hopfield network (Appendix D)]**
> In addition, we explain the robustness of the attention mechanism from a Hopfield network perspective using the connection of attention in transformers to the Hopfield network in the appendix.
>
> >Q2:\
> More general Transformer architectures should be explored in the paper. The authors mainly focus on ViTs, and the observations are mainly for ViTs. Recently, Swin Transformer gains more attention in computer vision tasks. The authors only discuss one type of Swin Transformer, i.e., Swin-S/4. It would be better to explore more types of Transformer and draw conclusions for the more general cases of Transformer.
>
> A2:\
> This is a great point! Due to space limitations, in the original submission, we put the main results of SOTA ViTs (including various Swin-T, ViT-SAM, DeiT) in the appendix and only selected some typical models in the main text, e.g., Swin-S/4. We refer to Table 7, Table 8, and Table 9 for more types of transformers. The appendix is probably not found by reviewers because we put it in a zip file. We have updated the paper with the appendix directly attached at the end for convenient checking.
>
> **References**\
> [1] Wang, Haohan, et al. "High-frequency component helps explain the generalization of convolutional neural networks." Proceedings of the IEEE/CVF Conference on Computer Vision and Pattern Recognition. 2020.\
> [2] Caro, Josue Ortega, et al. "Local Convolutions Cause an Implicit Bias towards High Frequency Adversarial Examples."

---

> ### Author Response · Authors · 2021-11-21
> **Response to Reviewer x14o (2)**
>
> >Q3:\
> The experiments should be improved. The authors only consider PGD attack and AutoAttack. These two types of attack methods are linear, i.e., add small perturbation on clean data. It would be better to conduct more experiments on other nonlinear attack methods, e.g., ADef [1], which applies small deformations to the clean data.\
> [1] ADef : an Iterative Algorithm to Construct Adversarial Deformations. ICLR 2019
>
> A3:\
> We further conducted experiments with ADef to check the robustness of vision transformers under ADef attack. We follow the default setting in their [GitHub released version](https://github.com/MashmallowWoR/AdversarialDeformation). The robust accuracy of ViT-S/16 is 12.4%, and the robust accuracy of {Vgg16, DenseNet, MobileNet, ResNet18} is {10.8%, 11.1%, 11.7%, 11.8%} respectively, which is in accordance with our conclusion. We use PGD and AutoAttack because they are among the most commonly used adversarial attacks for robustness evaluation. We have added this result in Appendix H of the revised version.
>
> >Q4:\
> In Table 1, the attack success rate (ASR) of Deit-S/16 is the best result, but the ASR of ViT-L/16 is 98.2. Is there a mistake in these two results?
>
> A4:\
> For Table 1, yes this is an incorrect annotation. The best-performing model that should be highlighted is ViT-L (98.2%) instead of DeiT-S (98.9%). Both of them are ViT-based and we show that ViT-based models tend to have lower ASR than others.

---

> ### Author Response · Authors · 2021-11-30
> **Request for further discussion**
>
> With the rebuttal period nearing a close, we sincerely request you to please let us know if you have further questions. To the best of our knowledge, we have addressed your concerns by concluding our contributions and referring to the appendix where we have actually put the experimental results you expected to see. It seems our appendix was not found because we put it in the zip file separately. Therefore, in the revised version, we directly attached the appendix at the end of the main paper. We appreciate a further discussion to quell concerns and revisit your rating, and more importantly, help us improve the paper overall.

---

### Official Review · Reviewer_7ATA · 2021-11-02

**Correctness:** 3
**Technical Novelty And Significance:** 2
**Empirical Novelty And Significance:** 2
**Recommendation:** 5
**Confidence:** 5

**Details Of Ethics Concerns:**


None.



**Main Review:**


(Negative) Throughout the paper, the authors claimed that this is the first paper to introduce an adversarial attack on ViT to examine the robustness of ViTs. But actually, until today, there have been a number of papers working on this topic, e.g., [r1-r4]. Especially, [r4] has been published in ICCV 2021. I understand that the authors might argue that this paper was submitted to Arxiv on 29 Mar 2021, which is earlier than some of the other papers. But, the authors should understand the following facts. First, please note that, [r4] was submitted to Arxiv on 26 Mar 2021, which is even earlier than the authors' paper. Moreover, [r4] has been published by ICCV 2021, and the authors' paper is unpublished yet. Second, because this paper is still not yet published and currently there has been a number of paper workings the same ideas of this paper, it would not be appropriate for the authors to overstate this is the first work in ViT robustness, considering many of the concurrent works provide more deep insights in this topic. Third, [r5] was the earliest paper studying the robustness of non-local attentive models among all of these works. Fourth, the insight of this paper is relatively shallow. There are not many valuable discoveries in this article, so the author keeps discussing the same thing from different angles. Fifth, I understand that the author wrote this paper many months ago, and they don't want to improve their paper because they think many "advance papers" are later than theirs. But I encourage the authors to improve their paper with more advanced insights to catch up with the recent developments. For example, the author could propose some practical solutions to ViT robustness. I also have some experience that I was the first to propose some techniques, but my paper was rejected. Then, the later paper working on a similar idea was published. I was frustrated. But I continued to improve my paper, and finally, my paper was accepted by another venue as an oral presentation.


[r1] Intriguing Properties of Vision Transformers

[r2] Vision Transformers are Robust Learners

[r3] RobustART: Benchmarking Robustness on Architecture Design and Training Techniques

[r4] Understanding Robustness of Transformers for Image Classification (ICCV 2021) 26 Mar 2021

[r5] Feature Denoising for Improving Adversarial Robustness.


(Negative) It would be good not to say that this work provides the first study on the robustness of vision transformers (ViTs) against adversarial perturbations. There have been other works discussing the ViT adversarial robustness. It would be good for the author to recognize the contribution of the community. It would be good for the authors to be humble.

(Negative) The following observations are natural and not new: "we find that ViTs possess better adversarial robustness when compared with convolutional neural networks (CNNs)."

(Negative) The following observations are natural and not new: "Features learned by ViTs contain less low-level information and are more generalizable, which contributes to superior robustness against adversarial perturbations." Existing papers found that ViTs do better than CNNs in adversarial robustness but do worse than CNNs in natural noise robustness.

(Negative) The following observations are complementary to the above observations. They thus are less informative: "Introducing convolutional or tokens-to-token blocks for learning low-level features in ViTs can improve classification accuracy but at the cost of adversarial robustness."

(Negative) The following observations are complementary to the above observations. So these observations are less informative. Increasing the proportion of transformers in the model structure (when the model consists of both transformer and CNN blocks) leads to better robustness. This makes readers feel that there are not many valuable discoveries in this article, so the author keeps discussing things from different angles.

(Negative) I'm afraid I have to disagree with the authors' following claim: "Pre-training on larger datasets does not significantly improve adversarial robustness though it is critical for training ViTs." I think the authors performed the wrong pre-training. Pre-training on the larger dataset should be robust pre-training, i.e., it should be adversarial training, but NOT vanilla pre-training. In my experiments, doing robust pre-training on a larger dataset significantly improves adversarial robustness.


(Negative) The following observations are too natural and contain little information: "ViTs are less sensitive to high-frequency perturbations than CNNs and
there is a high correlation between how well the model learns low-level features and its robustness against different frequency-based perturbations."

(Negative) The following descriptions need to be corrected: "it remains unclear on the robustness of ViT against adversarial perturbations, which is critical for safe and reliable deployment of many real-world applications," given that the literature has fully studied the robustness of ViT.


(Negative) The following descriptions need to be corrected: "In this work, we conduct the first study on examining the adversarial robustness of ViTs on image classification tasks and make comparisons with CNN baselines."


(Neutral) The following observation is natural and ok: " Using denoised randomized smoothing (Salman et al., 2020), ViTs attain significantly better-certified robustness than CNNs."

(Positive) The following description is correct and insightful: "short-term memory (LSTM) or CNN, with a theoretical explanation provided in Hsieh et al. (2019). However, due to the discrete nature of NLP models, these studies are focusing on discrete perturbations (e.g., word or character substitutions) which are very different from small and continuous perturbations in computer vision tasks."

(Negative) The following description needs to be modified: " To the best of our knowledge, this work is the first study that investigates the adversarial robustness (against small perturbations in the input pixel space) of transformers on computer vision tasks."

(Negative) The following description needs to be modified: "In the context of computer vision, the most relevant work is Alamri et al. (2020), which applies transformer encoder in the object detection task and reports better adversarial robustness."

(Negative) In the related work section, all the above-mentioned works should be discussed and compared to acknowledge the community's contribution.

(Positive) I am happy to see that the authors also evaluate the certified robustness of the models using randomized smoothing, where the robustness is evaluated as the certified radius,

(Negative) The following observations are too natural: " Adversarial training can be applied to train robust ViTs."


(Positive) The following finding is interesting and valuable: "We conjecture that ViT may need larger training data or longer training epochs to improve further its robust training performance, inspired by the fact that on natural training ViT is not able to perform well either without large-scale pre-training."


(Positive) The results in Table 4 are fascinating, valuable, and beneficial. I really like the results in this table.


(Negative) Please use text instead of figures to give a sub-caption to each subfigure in Figure 4. The current version is lossy.


**Summary Of The Paper:**


This paper studies the adversarial robustness of ViTs. There are several significant strengths and weaknesses in this paper. Especially, the novelty of this paper against one published paper is limited. It would be good to see my detailed comments below.



**Summary Of The Review:**


Balancing the strengths and weaknesses of the proposed method, I would like to recommend a rating of weak rejection for this paper.

---

> ### Author Response · Authors · 2021-11-21
> **Response to Reviewer 7ATA**
>
> We thank Reviewer 7ATA for the detailed comments. Your sharing of your personal experience and relentless efforts to improve your work is also very encouraging to us.  We agree that the study of robustness of ViTs  is a research topic that receives rapidly growing attention, and there have been some concurrent and followed up works in the literature. We discuss these concurrent works in Appendix G, acknowledge their contributions, and highlight the major differences to ours. We have also modified the sentences Reviewer 7ATA mentioned with proper expressions about the statement of our contribution and timeline.
>
> As Reviewer 7ATA has already found, we put this work on Arxiv in March. This work was started last year when ViT was proposed, so we think it should be more proper for us to say we are among the first works on this topic. **But we disagree with the assertion “they don't want to improve their paper because they think many ‘advance papers’ are later than theirs”. Actually, we continued our effort after the first submission.** For example, we added certified robustness analysis using denoised randomized smoothing, which compared the certified robustness of different models (we are unaware of any work that shows this analysis of certified robustness for ViT). The result shows ViT is also more robust than CNN in terms of certified robustness and this robustness should be due to the architecture design. We also supplement an explanation for the robustness of the attention mechanism from a Hopfield network perspective using the connection of attention in transformers to the Hopfield network in the appendix,and continuously add many SOTA ViT models that come up after our first submission to our study, e.g., Swin-Transformer [5], DeiT & Dist-DeiT [6], and SAM-ViT [7], which can also be found in the appendix.
>
> We also want to note that 3 out of 4 literature Reviewer 7ATA raised ([1, 2, 3]) indeed cited our work as references, and are months later than our work. [3] was published in September 2021, according to the terms of ICLR, we are not obliged to cite these works in current submission: “since our full paper deadline is October 5, if a paper was published (i.e., at a peer-reviewed venue) on or after June 5, 2021, authors are not required to compare their own work to that paper.” And [4] proposed to add non-local blocks for denoising, which is in the scope of adversarial training (though [4] did not mention our related result in Sec. 5.4). Their model is designed for CNNs as ViT came much later than this work. Although there have been many concurrent works studying the robustness of ViT, the distinction between different works are obvious. We have concluded the contribution of these concurrent work in Appendix G of the updated version.
>
> **References**\
> [1] ​​Naseer, Muzammal, et al. "Intriguing Properties of Vision Transformers." arXiv preprint arXiv:2105.10497 (2021).\
> [2] Paul, Sayak, and Pin-Yu Chen. "Vision transformers are robust learners." arXiv preprint arXiv:2105.07581 (2021).\
> [3] Tang, Shiyu, et al. "Robustart: Benchmarking robustness on architecture design and training techniques." arXiv preprint arXiv:2109.05211 (2021).\
> [4] Xie, Cihang, et al. "Feature denoising for improving adversarial robustness." Proceedings of the IEEE/CVF Conference on Computer Vision and Pattern Recognition. 2019.\
> [5] Liu, Ze, et al. "Swin transformer: Hierarchical vision transformer using shifted windows." arXiv preprint arXiv:2103.14030 (2021).\
> [6] Touvron, Hugo, et al. "Training data-efficient image transformers & distillation through attention." International Conference on Machine Learning. PMLR, 2021.\
> [7] Chen, Xiangning, Cho-Jui Hsieh, and Boqing Gong. "When Vision Transformers Outperform ResNets without Pretraining or Strong Data Augmentations." arXiv preprint arXiv:2106.01548 (2021).

---

> ### Author Response · Authors · 2021-11-30
> **Request for further discussion**
>
> With the rebuttal period nearing a close, we sincerely request you to please let us know if you have further questions. To the best of our knowledge, we have addressed your concerns about adding a discussion of concurrent works and modified the sentences you mentioned with proper expressions. We appreciate a further discussion to quell concerns and revisit your rating, and more importantly, help us improve the paper overall.

---

> ### Comment · Reviewer_7ATA · 2021-11-30
> **Post-rebuttal discussion**
>
>
> Thank you very much for the authors' reply. It can be seen that although the authors did not directly reply to my question here, they have already adopted some of my amendments in the revised version. Authors should directly respond to my other comments here; otherwise, AC and other reviewers will easily fail to notice my valuable comments.
>
> I understand and respect the "four-month rule." My main concern is that under the premise that many concurrent works have been published, the authors' papers were not published in time. Isn't it because the observations in this paper are not insightful enough?
>
> Specifically, some of the observations in this paper are relatively natural and informationless. For example, the following observations are redundant: increasing the proportion of transformers in the model structure (when the model consists of both transformer and CNN blocks) leads to better robustness. This makes readers feel that there are not many valuable discoveries in this article, so the author keeps discussing things from different angles. For another example, I think the authors performed the wrong pre-training. Pre-training on the larger dataset should be robust pre-training, i.e., it should be adversarial training, but NOT vanilla pre-training. In my experiments, doing robust pre-training on a larger dataset significantly improves adversarial robustness. For the third example, it is too natural to know that adversarial training can be applied to train robust ViTs.
>
> I respect the efforts of the authors, and I tend to give a moderate score.

---

> > ### Author Response · Authors · 2021-11-30
> > **Thank you for your feedback**
> >
> > We thank the reviewer for the valuable feedback. We now better understand how the reviewer positions the contributions of our work. We are glad that the reviewer found some of the suggestions were included in our revised version. We are also sorry to learn that the reviewer felt some comments were not directly responded to, which was definitely not our intention.
> >
> > Regarding concurrent works, we certainly don't agree with your assertion that "under the premise that many concurrent works have been published, the authors' papers were not published in time. Isn't it because the observations in this paper are not insightful enough?" First of all, as we discussed in the rebuttal, many mentioned works actually cited our work and acknowledged our contributions. Other than [r4] which is a concurrent work that we missed and later cited in the revised version, we do not think it's reasonable to cite follow-up papers citing our work, though we indeed provide a detailed discussion in Appendix G. Secondly, we do not think a paper under review and not published in time warrants diminishing values.
> >
> > Finally, regarding the findings from our work, we understand that the reviewer may feel some conclusions are "intuitive". But without rigorous experimental analysis like we did in our work, one cannot verify such statements. We indeed did not consider robust pre-training, but our main focus is in fact on the adversarial robustness of publicly available pre-trained models which do not use robust pre-training. To our best knowledge, our robustness analysis (such as Fig. 1) of ViT models is the largest-scale study to date in terms of the number of models included. Lastly, we would like to bring the reviewer's attention to other novel findings from our work. For example, we are the first study to show ViTs are certifiably more robust than CNNs using denoised smoothing (Sec. 5.3).

---

### Official Review · Reviewer_FTwY · 2021-11-03

**Correctness:** 3
**Technical Novelty And Significance:** 2
**Empirical Novelty And Significance:** 3
**Recommendation:** 5
**Confidence:** 4

**Details Of Ethics Concerns:**

N.A.

**Main Review:**

Strengths: \\
(1) As claimed, this work provides the first and comprehensive study on the robustness of ViT. \\
(2) This study provides some interesting (but not surprising) insight.
(3) The paper has a clear structure and is easy to follow.

Weaknesses: \\
(1) The insights of this work cannot be exploited for understanding or improving the adversarial robustness. This is the main reason that I believe this work might be below the ICLR bar. \\
(2) Why is ViT more robust than CNN? This work claims that ViTs have less low-level information and more generalizable. How do you define high-level and low-level? Why does high generalization contribute to superior robustness? Any proof or reference? It would be interesting if the authors can provide more concrete insight into the mechanism behind the reported phenomenon. \\


Some works [1-10] might be worth a check: \\
They do not affect my rating of this submission.

[1] Understanding and Improving Robustness of Vision Transformers through Patch-based Negative Augmentation
[2] Certified Patch Robustness via Smoothed Vision Transformers
[3] Adversarial Robustness Comparison of Vision Transformer and MLP-Mixer to CNNs
[4] Towards Transferable Adversarial Attacks on Vision Transformers
[5] On Improving Adversarial Transferability of Vision Transformers
[6] Reveal of Vision Transformers Robustness against Adversarial Attacks
[7] Intriguing Properties of Vision Transformers
[8] Vision Transformers are Robust Learners
[9] On the Robustness of Vision Transformers to Adversarial Examples
[10] Towards Robust Vision Transformer

**Summary Of The Paper:**

This paper performs a study on the adversarial robustness of vision transformers. It provides some insight on vision transformer from the robustness perspective.

**Summary Of The Review:**

Overall, this work has some merits but I expect more than that to get in for ICLR. Some investigation is rudimentary and is suggested to provide more deep insight.

---

> ### Author Response · Authors · 2021-11-21
> **Response to Reviewer FTwY**
>
> We thank Reviewer 1 for raising questions about the insights of understanding the adversarial robustness. We would like to supplement some supporting references and provide a more detailed analysis.
>
> >  Q1: The insights of this work cannot be exploited for understanding or improving the adversarial robustness. This is the main reason that I believe this work might be below the ICLR bar.
>
> A1:
>
> We would like to respectfully disagree with this comment and would like to re-iterate several insights of our work into understanding and improving the adversarial robustness of ViTs.
>
> In our feature visualization experiment, as shown in Figure 4, features learned by CNNs show obvious lines and edges which correspond to the high-frequency/low-level information in the images, while features learned by ViTs contain less such information. Therefore, it is natural for us to investigate the behavior of ViT against the high/low adversarial perturbations. In section 6.1, we show ViT is more robust than CNN especially against the high-frequency perturbations (i.e. adversarial perturbations with low-frequency components filtered out). But the ASR against low-frequency perturbations (i.e. adversarial perturbations with high-frequent components filtered out) is similar for CNNs and ViTs. Our frequency study shows that VITs have superior robustness against high-frequency perturbations, and the high-frequency components are one cause of adversarial examples as indicated by [1].
>
> We also provide a certified robustness analysis using denoised smoothing, which is to our best knowledge unexploited by other concurrent works. The denoised randomized smoothing applies to any model. The result suggests that ViT has better certified robustness which could be attributed to the model design.
>
> Our study also finds that using conv layers in ViT will weaken adversarial robustness (Sec. 6), which means the future design of ViT should consider a trade-off of adversarial robustness before adding new conv blocks into transformers.
>
> In addition, we explain the robustness of the attention mechanism from a Hopfield network perspective using the connection of attention in transformers to the Hopfield network in the appendix. As we put the appendix separately in a zip file and some reviewers may have not seen it, we updated our submission with the appendix attached at the end for more convenient checking.
>
> >Q2: Why is ViT more robust than CNN? This work claims that ViTs have less low-level information and more generalizable. How do you define high-level and low-level? Why does high generalization contribute to superior robustness? Any proof or reference?
>
> A2:
>
> Transformer uses self-attention so it tends to focus on high-level info, while CNNs use convolution and thus focus more on low-level. Our hybrid Conv-ViT verifies their effect on robustness. Our finding is also consistent with other (later) works, e.g., [8] further supports our conclusion by showing that the lack of convolutional operations in ViTs is responsible for this greater robustness to high-frequency attacks.
>
> [1] notices CNN’s ability in capturing the high-frequency components (HFC) of images, which are almost imperceptible to a human but critical for CNNs to boost accuracy (as shown in their Figure 2). As a result, the model uses such imperceptible HFC to make predictions, leading to generalization behaviors counter-intuitive to humans such as adversarial examples. [8] says their experiments further confirmed the robustness of ViT we observed may be due to the lack of convolutional layers, which leads to a bias towards high-frequency adversarial examples.
>
> > Q3:
> Some references
>
> A3:
>
> We also thank Reviewer 1 for pointing out some interesting works, which could help us rethink about the adversarial robustness of ViT from other perspectives. We are also delighted to find among these references, six of them [2,3,4,5,6,7] cited our work and recognized our contribution, and [3,5,8] conducted very interesting research based on or further confirmed our hypothesis. We have concluded these concurrent works and acknowledged their contributions in Appendix G of the updated version.
>
> **References**\
> [1] Wang, Haohan, et al. "High-frequency component helps explain the generalization of convolutional neural networks." \
> [2] Naseer, Muzammal, et al. "Intriguing Properties of Vision Transformers." \
> [3]  Naseer, Muzammal, et al. "On Improving Adversarial Transferability of Vision Transformers." \
> [4] Benz, Philipp, et al. "Adversarial Robustness Comparison of Vision Transformer and MLP-Mixer to CNNs." \
> [5] Aldahdooh, Ahmed, Wassim Hamidouche, and Olivier Deforges. "Reveal of Vision Transformers Robustness against Adversarial Attacks." \
> [6] Paul, Sayak, and Pin-Yu Chen. "Vision transformers are robust learners." \
> [7] Mao, Xiaofeng, et al. "Towards Robust Vision Transformer." \
> [8] Caro, Josue Ortega, et al. "Local Convolutions Cause an Implicit Bias towards High Frequency Adversarial Examples."

---

> > ### Comment · Reviewer_FTwY · 2021-11-27
> > **Thanks for the rebuttal**
> >
> > Thank you for the reply. First of all, as indicated before, the listed papers did not affect my rating, and the authors do not need to discuss them in this work. Since the main message of this work is to show that transformers are more robust than CNNs, I believe that it is important to give a deep analysis of why this is the case. Checking those works might inspire the authors to come up with a better explanation. One of my major concerns was that the main conclusion (transformers are more robust than CNNs) might be just caused by unfair training setups, thus requiring a convincing explanation of why this is the case. For example, a recent work [1] argues that "those conclusions (transformers are more robust than CNNs) are drawn from unfair experimental settings, where Transformers and CNNs are compared at different scales and are applied with distinct training frameworks". The explanation provided in the current version, such as high(low)-level is not fully convincing. On the other hand, I recognize that it is NOT easy to provide such a convincing explanation and the authors have probably tried their best for their current explanation. Considering the significance of the problem this work solves and the best effrot the authors have tried, I am neutral on the acceptance of this work even though my main concerns are not addressed by the reply.
> >
> > [1] Are Transformers More Robust Than CNNs?

---

> > > ### Author Response · Authors · 2021-11-28
> > > **Thanks for your reply**
> > >
> > > We thank Reviewer FTwY’s kindly providing more references for our consideration. We already summarized our main contributions and differences to existing works in  [the previous reply](https://openreview.net/forum?id=O0g6uPDLW7&noteId=Bhjv6dKtsk) and would like to note that [1] was made online on Nov 10th, which is even after the ICLR submission.
> > >
> > > We recognized the interesting idea in [1] to investigate the adversarial robustness of ViTs and CNNs under similar adversarial training schemes. In comparison, our work mainly focuses on the standard training setting and assumes models are trained in their commonly used practical settings. As found by [1] and shown in their Section 4.1, ViTs are more robust than CNNs in the standard setting which is in accordance with our results. They also found the training receipts for ViTs and CNNs can hardly be directly applied to each other. And in Table 2, they show that adversarially trained ResNet50 is less robust than adversarially trained DeiT, and attribute this to the use of GRLU activation functions which is inspiring. They show the better robustness of ViT against **patch-based attacks** is affected by the augmentations used during training, which is not in our current research scope.
> > >
> > > We are glad that Review FTwY is open and engaged with discussion, we hope our response had addressed the reviewer's concern and clarified the differences to [1], and that it will help the reviewer in making the final evaluation.
> > >
> > > [1] Are Transformers More Robust Than CNNs?

---

### Author Response · Authors · 2021-11-21
**General Response from Paper595 Authors**

We thank the valuable suggestions from reviewers. A new version of our submission has been uploaded with the changes marked in blue. There are three major changes:
1. Some expressions were revised with more detailed explanations to avoid confusion, e.g. we added  ​​the qualification of “without adversarial training” to the pre-training when concluding that the standard pre-training cannot obtain good adversarial robustness.
2. As pointed out by Reviewer 7ATA, our arxiv paper is among the first works to study the adversarial robustness of ViTs, and [1] is published on arxiv 3 days earlier than ours and is definitely a concurrent work. We have included it and other much recent related works (listed below) in Appendix G for discussing the concurrent and related works. Besides, although we newly added a discussion of these works, we want to note that other than [1], all the other 10 references reviewers mentioned were either
    * 1) released near the ICLR submission deadline. According to the guidelines of ICLR, we are not obliged to cite these works in the current submission: “since our full paper deadline is October 5, if a paper was published (i.e., at a peer-reviewed venue) on or after June 5, 2021 (e.g., [2,3,4,5,6,9,11]), authors are not required to compare their own work to that paper.”;
Or
    * 2) actually cited our work as reference or showed a follow-up study (e.g.[3,4,5,7,8,10,11]).

**References**\
[1] On the Robustness of Vision Transformers to Adversarial Examples\
[2] Certified Patch Robustness via Smoothed Vision Transformers \
[3] Adversarial Robustness Comparison of Vision Transformer and MLP-Mixer to CNNs \
[4] Towards Transferable Adversarial Attacks on Vision Transformers \
[5] On Improving Adversarial Transferability of Vision Transformers\
[6] Reveal of Vision Transformers Robustness against Adversarial Attacks \
[7] Intriguing Properties of Vision Transformers \
[8] Vision Transformers are Robust Learners \
[9] Understanding and Improving Robustness of Vision Transformers through Patch-based Negative Augmentation \
[10] Towards Robust Vision Transformer\
[11] Tang, Shiyu, et al. "Robustart: Benchmarking robustness on architecture design and training techniques." arXiv preprint arXiv:2109.05211 (2021).

---

### Decision · Program_Chairs · 2022-01-20

**Decision:**

Reject

**Comment:**

The paper studies the adversarial robustness of vision transformers. The authors conclude that vision transformers are generally more adversarially robust than the convolutional neural networks. Several interesting empirical conclusions are made for the robustness property of vision transformers. Sufficient empirical experiments are conducted. Overall, the paper is well-written, well-organized, and interesting. However, there are some concerns about the current version. (1) There are some concurrent works having similar empirical findings, which have been formally published and would weaken the interest of readers in the paper. (2) The reviews suggest that the authors use the insights from the paper to design more robust and effective vision transformers. The four reviewers have unanimous recommendations below the acceptance threshold. We therefore cannot recommend acceptance. However, we believe that by taking the comments, the next version would be a very strong paper.